# An Ecumenical Spirituality

**John Binns**

Institute for Orthodox Christian Studies, University of Cambridge, Cambridge CB3 9DF, UK; jb344@cam.ac.uk

**Abstract:** The modern ecumenical movement is a part of a wider ecumenism which expresses the universal character of the Christian faith. It is an approach to faith which is aware of the world-wide context of church life and the variety of the cultures and communities where it is practiced. The Orthodox Church of Ethiopia shows the importance of ecumenism because here we find a style of worship and theology which has taken a very different character from other parts of the church, especially in its relations with other faiths. Ecumenical faith recognises and welcomes difference and always seeks fresh ways of witness and proclamation. In a changing society, this ecumenical character of faith is an essential part of an effective mission and church life.

**Keywords:** *oikoumene*; Ethiopia; Judaism and Islam; science and faith; iconography; mission

## 1. The Meaning of Ecumenism

Ecumenism has a long history. The word ecumenism can be traced back to a set of Greek words relating to where people live. So, the noun *oikos* is a house, or a family or a race; the verb *oikeo* is inhabit or occupy. The phrase *oikoumene ge* is the inhabited land. Eratosthenes of Cyrene (276–194 BCE) was a polymath who was the chief librarian at Alexandria and a pioneer in the exploration of the natural world—being the first scientist to measure the circumference of the world. He coined the word geography, or writing about the earth, and used the term *oikoumene* to refer to the inhabited rather than uninhabited parts of the world. The evangelist Matthew, writing some time later, used that same term *oikoumene* also with that meaning of the universal and the world wide. A verse in his gospel looked ahead to the age to come when 'this good news of the kingdom will be proclaimed throughout the world (*oikoumene*) as a testimony to all the nations' (Mt 24.13). Then at the end of the gospel Jesus tells the apostles to 'go out and teach all nations' (Mt 28.19). For Matthew, the word *oikoumene* speaks of the missionary calling of the church to proclaim the gospel to all peoples and nations throughout the world.

Ecumenism is this awareness of this world wide character of the church. It describes a gospel message which cannot be restricted to one place or one people or one time. This gospel belongs in all cultures. It recognises and rejoices in difference. It is attentive and responsive to the character and conditions of the society in which it finds itself. It is faithful to the teaching and mission of Christ which is incarnated in the conditions of the place where it is lived out. In the Nicene creed we affirm our belief in a church which is not only one and holy and apostolic but also catholic and so is whole and complete wherever it is found. Ecumenism challenges us to live out our faith in this world wide, universal place where we share a common life within a faith community, scattered over the world yet gathered together by Christ. An ecumenical faith is part of this world, is shaped by it and speaks to it.

When they spoke of the *oikoumene*, the writers and readers of the New Testament would have had in their minds the Roman ruled regions around the Mediterranean. The Acts of the Apostles describes the spread of the gospel westwards through Asia Minor and to Rome and Spain beyond. There are however hints of other directions of travel for the gospel. While travelling the apostle Philip meets a court official of the candace of Ethiopia, in fact probably referring to an area of what is now Sudan (Acts 8.26-40),

and explains to him the true meaning of the prophecy of Isaiah. From this beginning the Christian faith extended further into the Horn of Africa. Later, in around 340, the kingdom of Ethiopia became one of the first nations to become Christian, along with Georgia, and the first Archbishop of Ethiopia was consecrated. The Ethiopian Orthodox Tawahedo Church—to give it its full title where Tawahedo means united referring to its affirmation of the one undivided nature of Christ—has remained the church of the nation. It remained Christian in the midst of the Islamic expansion across the Middle East, thrived alongside a sometimes aggressive missionary programme by both Catholics and Protestants, and is now the second largest national Orthodox Church after Russia, with some projections suggesting it will outstrip Russia in size by 2050 (Johnstone 2011, pp. 107–9). This vibrant church life shows the ecumenical dimension of the church in its distinctive patterns of worship and spirituality. Encounter with this African form of Christianity questions and challenges a western approach to Christianity, showing how a local culture can lead to a diversity of worship and practice of the faith. Experience of this church brings both challenges and renewal.

## 2. The Church of Ethiopia

My first experience of the Ethiopian Church was in 1993, and since then I have made regular visits. There have been personal discoveries showing how different ways of worship and styles of theological tradition can enrich and deepen my own faith. There have also been discoveries of fresh possibilities and understandings of the identity and life of the wider church. So, for me, sharing in worship with the Ethiopian Church is a vivid example of the richness and diverse character of the ecumenical universal church, both on the level of personal faith and on the level of ecclesiastical life.

As an illustration of how the Christian faith is lived in Ethiopia, here is an account of a celebration of the liturgy in a church in Gondar in the north of the country, taken from my diary of a three-month study visit to traditional church schools in 2008. It was part of a project to encounter and meet with the students of oral methods of church education. The Ethiopian Church education system has been claimed as the oldest in the world still being taught. Boys—it is mainly boys but there are some girls too—leave their homes and study at churches and monasteries following a syllabus which includes *zema* or church hymnody, *qene*, a theological study using traditional forms of improvised poetry, *akwakwam* or the ritualised movements and music used in liturgy, and *meshaf* or the interpretation of Biblical and theological texts. I had spent time getting to know teachers and students at the traditional church schools in the city of Gondar, a centre of this ancient tradition of scholarship. As I gradually became accustomed to the way students were taught and how they understood their faith, so my own faith was disturbed and changed (Alemayehu 1971; Binns 2013, 2017, pp. 159–94).

So here is my description of the celebration of the liturgy.

'I arrived at the church on Saturday evening soon after the sun had set and as darkness was falling. We went to the nun's house—a corrugated iron shack. In one corner she had built a fire and was making a kind of bread called *sanbatkitta* which would be given out to everyone after the service. Her pile of flat loaves was growing'.

'At about 7.45 we went into the church. The church students of *qene* and *akwakwam* were gathering in the *qene mahlet* (or place of music at the west end of the church). The head of the church, clergy and teachers sat against the west wall and the students gathered around them so that they formed a square. Drums, *maqomiya* (prayer sticks) and *sistrums* (a kind of rhythmic instrument used in worship) were distributed, with a friendly rivalry and arguments as each tried to make his neighbour take the better made stick or instrument. I was given my stick and sistrum and tried to join in. The chants were long and repetitive with each lasting half an hour or more. The rhythm of the sistrums was confusing and I had to watch hard to keep in time. After a couple of hours there was a short break with prayers. All the students knew what they were doing and there were no books. The students were small in height and so looked younger than they probably were. One student was crippled,

two others were blind and were led by friends. The teachers were dressed in white for the worship. The head of the church sat in the middle presiding and remained motionless through the night. By 2 a.m. the students were tired, some fell asleep as they stood leaning on the prayer sticks, others curled up in a corner for a short rest. About 3 a.m. the clergy moved to a reading desk and sang the *kidan* a service of preparation for the *qiddase* or eucharist. At 5 a.m. the students stood to sing the last chant with enthusiasm, and when it finished, they returned to their huts to rest while the clergy sang the *qiddase*'.

'I stayed while the two priests and three deacons sang the liturgy. I watched as they came out from the *maqdas* or sanctuary and chanted the five readings from the Bible. Then as the priest washed his hands after the recitation of the creed, we all bowed to each other at the Peace. There were about thirty people in the church, some young children who received holy communion, the priest shielding the holy bread from their sight with his hand as he placed it in their mouths'.

'The worship had taken place on several levels. In the *maqdas* or sanctuary, concealed from the sight of the people outside but audible through a loudspeaker, the holy action of the *qiddase* is performed. In the *qeddist* or chancel the prayers of the *qidan* and the *saatat* or hours are sung. In the *qene mahlet* the hymns are sung and performed through the night. I knew that outside in the church compound people are standing reading the psalter or sitting or sleeping on the ground'.

'Around 9 a.m. the *qiddase* is over, and I can go out into the bright morning sun, I am stiff and aching but am amazed at the sight which greets me. The compound which had been almost empty when we entered the church is now crowded with people all wrapped in their white robes. It is a sight of tranquillity and purity, with the light of the sun, the swaying of the green trees, the thick long grass with splashes of colour as the weaver birds dart from bush to bush. After a time of preaching with several sermons by various clergy, everyone sits in rows and the deacons bring the *sanbatkitta*, the thick dark bread made the day before, and distribute large pieces to everyone present. People talk, laugh and, eventually and reluctantly, leave'.

'On my way home, after well over twelve hours in church, I stop at the first bar I come to, and order a strong black coffee. I am exhausted'.

### 3. The Place of Ethiopia

I had thought that I was familiar with different kinds of church life. I have been a priest of the Church of England for forty years; during training I spent a memorable six months working at the Serbian Orthodox theological faculty in Belgrade in the 1970s and have valued relationships with Orthodox as well as Anglican communities through the Fellowship of St Alban and St Sergius, during which I made several visits to Moscow, and the Institute for Orthodox Christian Studies; then I researched the writings of the desert fathers of Palestine which led to a period at St Georges College Jerusalem, and I have visited several other national churches during further research and writing. However, I was not prepared for the unexpected and exciting experience of sharing the life of the church in Ethiopia. It showed me again the way that the church forms its identity and carries out its mission as it lives, teaches and ministers in the conditions of the society in which it is located.

To understand the unique character of the Ethiopian Church, we need to look at a map. Ethiopia—including modern Eritrea—is located in the Horn of Africa. It is linked by trade routes with the interior of sub-Saharan Africa, looks across the Red Sea to Arabia, is connected by sea and land routes to Israel/Palestine, and it is not far by sea to India and East Asia beyond. This location, at a point between Europe, Africa and Asia, shows how the ecumenical church can develop a wide diversity of worship and practice. This is show clearly by its Semitic culture, shared with the other great monotheistic faiths of Judaism and Islam, which leads to a mutual coexistence.

## 4. Judaism

Historians have noted the influence of Judaism religious and cultural practices. In my night worship in the church at Gondar, I was aware of the way that layout of the church mirrors that of the temple at Jerusalem with its threefold division, which in Ethiopia is usually a series of three concentric circles, with the *maqdas* or holy of holies in the centre, surrounded by the circular *qiddist* where people who will receive communion stand, then the next circle is the *qene mahlet* where the debtera perform the church's hymnody, then there is the compound or outer court where the congregation sit or stand. At the centre of every church is the tabot – which is a representation of the Ark of the Covenant which resides, we are told, in the Church of Mary of Zion at Axum (Ullendorff 1968).

I recall being reminded by a deacon in Addis Ababa that 'our Ethiopian church is very old, it was founded many centuries before the birth of Christ'. He was thinking of the great epic of Ethiopian tradition—the *Kebra Negast* or Glory of Kings—which includes a fuller account than in the Bible of the visit of the Queen of Sheba to Solomon, when, after a lengthy encounter, she gave birth to a son who became king of Ethiopia and brought the Ark of the Covenant from Jerusalem to Ethiopia, where it is still cared for. It is a lengthy and involved account—drawing on Biblical and Quranic traditions of the Queen of Sheba, or Makeda as she is known in Ethiopia—and shows the Ethiopian vocation of being chosen by God and with a special place in the divine dispensation (Budge 2000).

Its history developed alongside those of both Judaism and Islam. Many practices are shared with Judaism. Boys are circumcised, the Old Testament dietary regulations are observed, the liturgy is celebrated on Saturday as well as Sunday, the *debtera* (who I met in Gondar) are often seen as inheritors of the Levites of the Old Testament. There has been extensive debate among scholars about when and how these traditions reached Ethiopia, but it is not necessary to identify a date or period of history. Rather it points to a shared culture and expression of faith.

## 5. Islam

There has also been a close relationship with Islam—after all, there are only a few hundred kilometres from the Christian capital of Axum to the Islamic heartland at Mecca. Islamic traditions of the life of the prophet tell how, when his followers were driven out of Mecca in 615 during the exodus known as the hijra, Muhammad sent eighty-two followers to the Christian king of Ethiopia because the king 'will not tolerate injustice and it is a friendly country'. They reported that 'the Negus (king) gave us a kind reception and we safely practised our religion and worshipped God and suffered no wrong in word or deed'. The king wanted to know more about what they taught and summoned his bishops to listen. When they heard, we are told, the king wept until his beard was wet and the bishops wept until their scrolls were wet, and the king said, 'of a truth this and what Jesus brought have come from the same niche'. (Guillaume 1955, pp. 150–55). Later, when the Arab armies were spreading across the Middle East, Muhammad told them, 'leave the Abyssinians alone so long as they do not take the offensive'. Since then, Muslims have formed a significant minority in Ethiopia living—usually—peacefully within the Christian kingdom.

While not given so much attention in the histories, the Ethiopian church life is also influenced by traditions coming from sub-Saharan Africa. As in much traditional African religion, the features of the natural world—trees, hill tops, streams and caves—have a spiritual dimension, making present the high and distant god of the sky here on earth. The spirits, too, need to be respected and recognised as present in human life, with evil *djinn* avoided and the more beneficent *zar* lived with.

So, a relationship between the three faiths of Christianity, Judaism and Islam grew through their shared geography, culture and religious customs. The three faiths have co-existed harmoniously in one national community. There have been periods of violence and conflict in Ethiopia, most notably between 1529 and 1543, when the emir of the sultanate of Adal, Mahmad ibn Ibrahim al Ghazi, devastated the churches of the Christian highlands, in a military campaign. But through most of its history, Ethiopia has been a place where

religions have lived harmoniously and peacefully together (Ehrlich 2007). The words and ideas of anti-Semitism and Islamophobia which point to deep rooted and tragic wounds within European society are absent from Ethiopian church life.

My encounter with the life of the church in Ethiopia showed me a rich tradition of spirituality and theology. There was so much which was unfamiliar and exciting. There was the oral method of teaching and tradition of theology which has its own vitality and freshness, different and sometimes absent from the more rigid reliance on fixed written texts (Vansina 1985); there was the worship in the holy places, which were found in beautiful and often remote mountain tops or deep caves, rather than in meeting places for worship in town centres; there was the strict fasting discipline, with over 250 days a year designated as fast days, which can lead to an identification with the gospel message. Then there were the different relationships with people of other faiths, especially the monotheistic faiths of Judaism and Islam—so unlike the suspicion and hostility of many churches. This all showed me a new depth and variety of the faith of the church, and the way that the Christian faith can find alternative ways of living in the kingdom of God which are rooted in the culture and traditions of the community.

So, my time spent in Ethiopia led me to reflect again on my own faith. I had my psalter and Bible with me so that I could continue with my regular pattern of Morning and Evening Prayer, but it was unsettling and awkward to return from the intense and demanding experience of being present at an Ethiopian liturgy to turn to my Anglican forms of prayer. It was one example of that experience of entering into the experience of faith and worship of another tradition—even if only in a temporary and superficial way—and recognising the surprising, intense, emotional quality of the experience, made more engaging by the hospitality and generous friendship of my hosts. This was, for me, an ecumenical encounter, as it made me more aware of the global character of the church as it is given its own distinctive shape and life as the person and presence of Christ comes to dwell in the hearts of believers.

## 6. An Ecumenical Faith

Difference can be difficult for the church to accept and cherish. Histories of divisions and schisms between church communities and conflicts between religions through the ages are all too familiar. Today we experience controversy over different understandings of gender and of sexuality. Hopes of a serene and untroubled faith undisturbed by doubt or unexpected surprises are broken by these disturbing challenges of difference.

While faith may not lead to tranquillity and calmness, it gives us conviction and confidence so we can witness to the salvation brought by Christ. If our encounter with other church communities or forms of worship is a genuine and honest meeting in which we explore the unfamiliar and discover fresh insights, then faith should be open to being tested and renewed, as mine was during those nights of prayer in the Ethiopian churches.

I have been given a manuscript of a set of writings by the Russian Metropolitan Anthony of Sourozh (1914–2003). Metropolitan Anthony trained as a doctor and this scientific background gave him a distinctive approach to faith, as he has described in a series of writings. His biography shows how his teaching was shaped by the convergence of the traditions of the theology of the church fathers, Russian spirituality and thinking, while lived out in a western, scientific society. In the course of a lengthy series of talks on the Creed given over several years, he set out a broad and generous vision of faith. Metropolitan Anthony shows us that a mature faith is bound to be provisional and uncertain. He describes the way a scientist works, conducting experiments, collecting data, constructing a model, and then re-examining the data, showing how the model falls short and then working on a more complete and accurate way of understanding.

'When a scientist has made or has inherited a certain knowledge and added to it his own discoveries, then he tries to hold together all the elements of knowledge by joining them together, by adjusting them to one another, by building what the scientists call a model, that is a vision of the world or of a particular phenomenon in the world in which

all the elements of knowledge fit. No scientist believes that this model is in all truth an adequate picture either of the phenomena or of the world, but it allows him to hold together, as it were, in one hand all the elements of the problem'.

'When a scientist has built up a model which did not exist before her and which holds together facts which could not be joined together before, then she probably has a legitimate sense of satisfaction. But if she is a true scientist, she will then try to find out whether there is a flaw in her construction, and if there isn't, she will eagerly begin to try and find the facts which cannot fit in her model, because the advance of science is made of the discovery of such facts that explode the existing model and force the scientists individually or collectively to build another model which is more comprehensive, more refined and nearer a true image of what really is. There is an exhilarating element in scientific doubt, in questioning the model which one has held to be true or built oneself. There is a hope that it will be given to the very person who worked out this model to find the fact that will break it, smash it to pieces. At no moment does a scientist doubt the objective reality which she is trying to express by her model. What she doubts is the model, her vision, her understanding, her ability to express reality in a way which is accessible to the human mind'.

'What happens to the believer very often is that he has held a model either of God or of the world, and this model is incomplete, and often primitive. At a certain moment this model becomes untenable. And at that moment instead of rejoicing the believer is in agony, because it seems to him, wrongly, that if his model is wrong and goes, then perhaps the reality that stood behind it must go and so is untrue. But as believers we have to grasp that doubt must be treated in a different way. We have to realise that doubt means a fracturing of a limited kind of certainty. Instead of being monolithic, whole, integral, we begin to see that there may be an alternative to what we thought was the only possibility. This condition normally should grow out of an ever-increasing experience of life in its entirety, of God, of man, of oneself and of the world that surrounds us. The images which are given us when we are children cannot be carried throughout a life without change, because they were not calculated to confront the other realities which were outside of a child's experience' (Unpublished manuscript).

This account shows us how our faith must be open to what is new if it is to be an ecumenical faith. It will be rooted in a tradition and experience of faith where we have encountered the risen Christ. But it will also be directed outwards and open to insights from the rich life of other communities. It will always be a seeking to approach and grasp the gifts of the Spirit as it works in new and unfamiliar ways.

## 7. Rebooted Ecumenism

Within the long tradition of ecumenical faith, there have been periods when ecumenism has taken a specific form and named itself as an Ecumenical Movement. These have emerged when political and social changes in society have led to an awareness of the urgency and priority of expressing and living the universal context and identity of faith. So, the church has reflected on and affirmed this universal identity and unity of mission.

In the fourth century, the Emperor Constantine ended the period of persecution of the church with the Edict of Milan (313 CE). Now that the church had a recognised place in the Empire, it needed to define its beliefs and overcome division. A series of councils met beginning with the Council of Nicaea in 325 and ending with a second Council of Nicaea in 787. These councils are called Ecumenical Councils because of their bringing together all parts of the church in the formulation of faith—although different communities give ecumenical authority to a varying set of councils. Then, many centuries later, after the end of the First World War, when nations were seeking to build a new international order and the League of Nations was being set up, an encyclical letter was sent by Patriarch Germanos of Constantinople in 1920 which he addressed to 'all the churches of Christ wherever they may be ...' calling for a world-wide fellowship of churches. This was the start of the modern ecumenical movement and led to the decision to set up a Council of Churches

taken in 1937, although its formation was delayed by ten years as a result of World War II. Its website states that its membership now consists of 352 churches representing over 580 million Christians. It continues to meet, sharing gifts of its churches and seeking to respond to the needs, challenges and sometimes persecutions which its members face.

A rebooted ecumenism, an ecumenism for our age, is a response to the character and the demands of the society in which we live. It challenges faith to become rooted in the society in which it lives, to be aware of the variety of ways that faith is lived and expressed, and to seek to live out faith in ways which enable its members to deepen and enrich their own faith, and enable the church to live out its mission to the world. In a world where travel is fast and convenient, where communication is instant and easy, where both ecclesiastical and faith communities live and work together, then that readiness to embrace and rejoice in diversity which is true ecumenism must be a continuing tradition of faith. Our rebooted ecumenist is constantly challenged and disturbed by encounters with other Christian traditions. But as well as being disturbed, she is also enriched and renewed by these fresh insights. She will also turn towards the world, recognising that the Christian Church has to live in and witness to that changing community in which it lives.

### 8. An Ecumenical Mission

Back in England, after those visits to Ethiopia, I continue to worship but now in my own home church communities. I live a few miles from the Anglican cathedral at Lichfield, a little north of Birmingham, and I value its openness to the city around it and its inclusion of different expressions.

In recent years, I have a watched a set of huge icons being painted and hung in the nave of the cathedral. There is the Annunciation in two panels on either side of the nave, and a huge hanging two-sided icon with the crucifixion on one face and the risen Christ on the other. It was painted (I avoid the practice of referring to icons as 'written', since the Greek *grapho*—from which iconography is derived—simply means to make a mark on paper and so can apply to painting as well as to writing, and so we can keep the obvious idea of icons being painted) by members of the Bethlehem Icon Centre located in the town of Christ's birth under the protection of the Greek Melkite Patriarch of Antioch and staffed by painters of different churches across the Middle East. They set up a studio in the transept of the cathedral throughout a summer and invited members of the congregation to see the icons being created. It was dedicated and hung at the centre of the cathedral in September 2018, watching over those who come in and showing God's love for all who worship.

Then in the apse to the east of the High Altar is the newly refurbished shrine of our founder St Chad. It contains a relic of the body of the saint which was presented to the cathedral by the Roman Catholic diocese of Birmingham in a moving ceremony in 2022. In this dedication service, the Roman Catholic and Anglican Churches expressed their shared faith and mission. The presence of both the icons—painted by artists from Bethlehem—and the relic of the saint—given by the Roman Catholics of Birmingham—express a shared mission to the people of the Midlands and of Birmingham. The east end is now a remodelled space as a shrine of silence and a destination of pilgrimage.

In the hot summer of 2022 and then again in 2023, the cathedral and city council built a sea-side resort outside the great west doors of the cathedral for children and their families to relax in. There was a beach with a large area of sand in which children could come and play, with deck chairs for parents and friends to rest and meet, an ice cream stall to offer refreshment, and a boat which was also a play area. During the school holidays, the cathedral became a place for the children of the city to come and play, and for adults to rest and enjoy warm sunshine. A visit to the cathedral in the summer has become a time of joy and celebration, with people from the town relaxing at the doors of the cathedral.

Lichfield Cathedral, with its three spires, dominates the city skyline. It is home to many groups and events. The icons, the shrine and the beach are just some examples of the variety and creativity of the life of the cathedral. It is an example of how one community has tried to open its life to other traditions of faith and welcome to its doors that community

where it is set. This, for me, is an example of an ecumenical spirituality. It has embraced and explored the riches of a different tradition in the placing of icons at the centre of the building. It has shared with a neighbouring church in the setting up of Chad's shrine. It has worked with partners in the wider community in opening up to welcome all who come. This has the marks of an ecumenical spirituality—ecumenical because it is open to and engaging with the wider community or *oikoumene*; ecumenical because it has listened to and learned from traditions of other parts of the church; ecumenical because it is rooted in and relates to the society around it—witnessing to the presence of the one God incarnate in his world. When I visit the cathedral, it shows me how the relationship with the city and the life of different Christian communities can help us grow together to become a whole church and enjoy our faith in many ways.

## 9. Conclusions

Ecumenism is the often unsettling and constant reminder that each of our local churches and communions are not the only manifestations of truth, but they belong within the *oikoumene*, that world-wide community of faith. It affirms that we as the church are entrusted with a message of salvation for the world. The ecumenical movement is that tradition which responds to the gift and challenge of the universality of the church.

The church today proclaims and lives its faith in a rapidly changing world with a new awareness of the variety of nationalities, cultures and faiths. This leads to a call for a new, rebooted ecumenism, in the faith and life of us all. While the ecumenical movements have given life to the church at moments of special need, the true ecumenism is a way our faith in a universal or ecumenical church is lived. An ecumenical faith grows out of our faith in Christ given through the gift of the Spirit at our baptism and shown in our worship and discipleship. It leads us to recognise that this membership is of a world-wide church, an *oikoumene* which embraces all inhabitants of this universal community. The diversity of this multi-racial and multi-ethnic community, set within a globalised political, economic and social process, forms the backdrop of our discipleship of Christ. Ecumenism is the way that become the church. It leads us to rethink our priorities and our faith. It is always present and so is always in the course of being re-booted. It leads us together into the fulfilment and joys of the Kingdom of God.

**Funding:** This research received no external funding.

**Conflicts of Interest:** The author declares no conflict of interest.

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
