# Peer review of "An Ecumenical Spirituality"

_religions, doi:10.3390/rel14101238_

Round 1

Reviewer 1 Report

In this paper, the author draws attention to the faith life of the Orthodox Church of Ethiopia. He calls attention to his experience of the liturgy in Gondar, which he describes, places the church of Ethiopia in historical perspective, and shows how this church has a special relationship both with Judaism and Islam, with which it generally coexists peacefully. The author links these experiences to a particular conception of ecumenism, which for him is synonymous with universalism, and uses views of the Russian metropolitan Anthony of Sourozh to argue for an ecumenical experience of faith: an experience of faith that continually renews itself and allows itself to be nourished by the various faith experiences of the one Christian faith worldwide. He gives as an example some particular, recent aspects of the faith life of Lichfield Anglican Cathedral.

This is not a successful scholarly paper. A clear question statement is lacking. The term ecumenism is interpreted in an unusual way. The description of the liturgy in Gondar raises many questions that are not discussed. Quotes are given, but from what? Nor is the extensive quotation from Metropolitan Anthony of Sourozh verifiable, since the text has not been published. There are no peers with whom the author has a conversation. There are no insights he challenges or even discusses. Clear argumentation is lacking. The contribution is mostly of a personal nature, and it could be that this person is so well-known and influential that this gives the paper a certain value, but the content by itself does not justify the personal nature of the text.

In addition, the text requires serious editing, as there are sentences that do not flow, words that are missing or need to be deleted, and it is full of unnecessary spaces. Take just the incomprehensible first sentence of the summary: what is meant here?

Author Response

Thanks for these comments.  They have led me to revise the paper. 

Its not intended as an academic contribution to a discussion on the modern ecumenical movement.  Rather it is a reflection on the relation of the church with society, and suggestions as to a spirituality and approach to mission which takes account of the worldwide ecumenical dimension of faith.  

I hope this provides a clearer statement of the p;urpose of the paper.

I think that its appropriate to include unpublished material such as Metropolitan Anthony's account of faith, as is often done in academic work.

Reviewer 2 Report

This is an eccentric and personal research report posing some challenges for peer review:

1. it is easy for this reviewer to identify the author as the Rev. Dr. John Binns of Cambridge

2. it mostly takes a cultural anthropological approach, including Participant Observation and a lengthy quotation from the author’s 2008 field notes (referred to as “diary”), and other very specific and personal references

3. there is no research design or rigorous methodology; it is almost a “memoir”

4. discussion of ecumenism based almost exclusively on Orthodox Christian expressions and the perspective of one Anglican priest is limited to a tiny part of the historic and global reality

And yet…

Perhaps a brief personal account that omits reference to the 1910 Edinburgh Conference, misdates the 1948 establishment of the World Council of Churches and minimizes its extensive day-to-day work around the world, and seems to ignore Vatican II can serve as a tiny, specific addition to the record of ecumenical activity. It certainly provides information about Ethiopia and Orthodox Christianity that would be new to most European- and North-American-centered ecumenists.

So I say, “Publish.”

But first “Proofread and edit.”

I caught annoying “typos” and errors of grammar and syntax in lines:

24

97

122-124

148

206

209

224

227

230-233290

293

321

333

362-363

I have some further specific comments:

Lines 31-32     “It recognizes and rejoices in difference.”

Yes.

Ideally.

Up to a point.

As an ecumenically-engaged church leader in Canada, I have seen and participated in tremendous ecumenical cooperation in lobbying with governments, sponsoring refugees, feeding the hungry and many, many local ministries. I have also seen a distinct lack of rejoicing when denominations have taken different public positions on issues of sexual expression, gender identity and reproduction resources.

Perhaps an accurate phrase would begin, “At its best, ecumenism…” or “Ideally…”

Line 61

It isn’t clear how Alemayehu, 1971 relates to the preceding.

Line 111

“forty years”

That is perhaps no longer a 2008 reference? That would be the author’s 2023 statistic?

Lines 245-277

The language of a source document can be at odds with editorial policies sensitive to inclusive language and language of justice. Use male references for “scientist” enough and one is less likely to expect to encounter a female scientist. I felt battered by the male usages in this section, and I self-identify as male and use he/him.

See the list of typos in Comments section.

And note reference to patriarchal language.

Author Response

I am grateful for these comments and have revised the article.  Among revisions I have ensured that Metropolitan Anthony's words are gender-inclusive.  As you point out it does not discuss the modern ecumenical movement, but shows how ecumenism is an aspect of faith and spirituality and should be reflected in our church life.  Looking at the challenges facing the church, including declining numbers, this seems to me a relevant contribution.  

Reviewer 3 Report

The Introduction offers a refreshing and helpful etymological reminder of the nature of what ecumenism means, contextualized by scripture. 

Sections revisiting the religious history and traditions of Ethiopia were also quite welcome. I appreciated how the Orthodox priest/scientist was used to introduce the importance of models and faith before returning to a discussion of the importance of community as a holding space for the birth of a deeper and more expansive/accepting kind of faith. 

L. 321 has a typo. Spacing needs corrected throughout--one space after periods, rather than two.

Overall, this does a great job of moving from personal reflection to a broad historical awareness of Christianity, establishing an appropriate case for ways of orienting toward an enriched future Christian tradition.

Author Response

Thanks for these comments.  I have revised the article to bring out these themes more clearly.  

Round 2

Reviewer 1 Report

The original version of this paper was more than superficially edited. This I recognize and appreciate. The abstract is definitely improved, and this leads to a better understanding of the rather uncommon concept of ecumenism the author employs: the universality of the Christian faith. The author also explains that he did not intend to write a scholarly paper, which is what most of my criticism came down to. So now the question is for the editor: does the journal admit of such a contribution of a more personal nature, or not? I assumed Religions to be a scholarly, academic journal; if so, my advice would be to reject. If it is meant to be broader than that, this contribution can be published (after minor revision).

There are still a number of typo's (escribes in stead of describes; are message in stead of a message; an court official in stead of a court official; A The words in stead of The words; if icons in stead of of icons).